# Population sizes, HIV prevalence, and HIV prevention among men who paid for sex in sub-Saharan Africa (2000–2020): A meta-analysis of 87 population-based surveys

Caroline Hodgins[1], James Stannah[2], Salome Kuchukhidze[2], Lycias Zembe[3], Jeffrey W. Eaton[4], Marie-Claude Boily[4], Mathieu Maheu-Giroux[2]*

1 Department of Microbiology and Immunology, McGill University, Montréal, Québec, Canada, 2 Department of Epidemiology and Biostatistics, School of Population and Global Health, McGill University, Montréal, Québec, Canada, 3 Joint United Nations Programme on HIV/AIDS, Geneva, Switzerland, 4 MRC Centre for Global Infectious Disease Analysis, School of Public Health, Imperial College London, London, United Kingdom

* mathieu.maheu-giroux@mcgill.ca

**Data Availability Statement:** The data underlying the results presented in the study are available

## Abstract

### Background

Key populations, including sex workers, are at high risk of HIV acquisition and transmission. Men who pay for sex can contribute to HIV transmission through sexual relationships with both sex workers and their other partners. To characterize the population of men who pay for sex in sub-Saharan Africa (SSA), we analyzed population size, HIV prevalence, and use of HIV prevention and treatment.

### Methods and findings

We performed random-effects meta-analyses of population-based surveys conducted in SSA from 2000 to 2020 with information on paid sex by men. We extracted population size, lifetime number of sexual partners, condom use, HIV prevalence, HIV testing, antiretroviral (ARV) use, and viral load suppression (VLS) among sexually active men. We pooled by regions and time periods, and assessed time trends using meta-regressions. We included 87 surveys, totaling over 368,000 male respondents (15–54 years old), from 35 countries representing 95% of men in SSA. Eight percent (95% CI 6%–10%; number of surveys [$N_s$] = 87) of sexually active men reported ever paying for sex. Condom use at last paid sex increased over time and was 68% (95% CI 64%–71%; $N_s$ = 61) in surveys conducted from 2010 onwards. Men who paid for sex had higher HIV prevalence (prevalence ratio [PR] = 1.50; 95% CI 1.31–1.72; $N_s$ = 52) and were more likely to have ever tested for HIV (PR = 1.14; 95% CI 1.06–1.24; $N_s$ = 81) than men who had not paid for sex. Men living with HIV who paid for sex had similar levels of lifetime HIV testing (PR = 0.96; 95% CI 0.88–1.05; $N_s$ = 18), ARV use (PR = 1.01; 95% CI 0.86–1.18; $N_s$ = 8), and VLS (PR = 1.00; 95% CI 0.86–1.17; $N_s$ = 9) as those living with HIV who did not pay for sex. Study limitations include a

from: Demographic and Health Surveys (DHS) and AIDS Indicator Surveys (AIS) (https://dhsprogram.com/Data/), Population-based HIV Impact Assessment (PHIA; https://phia-data.icap.columbia.edu), Kenya AIDS Indicator Survey (KAIS; www.knbs.or.ke), and South Africa National HIV Prevalence, Incidence, Behavior and Communication Survey (SABSSM; http://ghdx.healthdata.org/series/south-africa-national-hiv-prevalence-incidence-behavior-and-communication-survey-sabssm).

**Funding:** MM-G's research is supported by a Tier 2 Canada Research Chair in Population Health Modeling (https://webapps.cihr-irsc.gc.ca/decisions/p/project_details.html?appIId=415376&lang=en). This study was funded by the Canadian Institutes of Health Research (https://webapps.cihr-irsc.gc.ca/decisions/p/project_details.html?appIId=420347&lang=en). CH was supported through a *McGill Undergraduate Global Health Scholars Award*. JS received a doctoral award from the *Fonds de recherche du Québec— Santé* (FRQS). The funders had no role in study design, data collection and analysis, decision to publish, or preparation of the manuscript.

**Competing interests:** I have read the journal's policy and the authors of this manuscript have the following competing interests: MM-G reports funding from UNAIDS, the World Health Organization, the Institut national d'excellence en santé et services sociaux, the Institut national de santé publique du Québec, and an investigator-initiated grant from Gilead Sciences Inc., outside the submitted work. JS is supported by a doctoral award from the Fonds de recherche du Quebec-Sante (FRQS). JWE reports grants from UNAIDS, the Bill & Melinda Gates Foundation, US National Institutes of Health, and the World Health Organization and personal fees from the World Health Organization, all outside of the submitted work. All other authors declare no competing interests.

**Abbreviations:** AIS, AIDS Indicator Survey; ARV, antiretroviral; DHS, Demographic and Health Surveys; KAIS, Kenya AIDS Indicator Survey; MICS, Multiple Indicator Cluster Surveys; PHIA, Population-based HIV Impact Assessment; PR, prevalence ratio; SABSSM, South Africa National HIV Prevalence, Incidence, Behaviour and Communication Survey; SSA, sub-Saharan Africa; UNAIDS, Joint United Nations Programme on HIV/AIDS; VLS, viral load suppression.

reliance on self-report of sensitive behaviors and the small number of surveys with information on ARV use and VLS.

## Conclusions

Paying for sex is prevalent, and men who ever paid for sex were 50% more likely to be living with HIV compared to other men in these 35 countries. Further prevention efforts are needed for this vulnerable population, including improved access to HIV testing and condom use initiatives. Men who pay for sex should be recognized as a priority population for HIV prevention.

## Author summary

### Why was this study done?

- Key populations, including sex workers, are at a high risk of HIV acquisition and transmission.

- Clients of sex workers often have central roles in HIV transmission networks: they have sexual contacts with both sex workers and other partners.

- Despite their vulnerability to HIV acquisition and transmission, male clients of sex workers are not formally recognized as a key population.

- Characterizing HIV epidemiology among men who pay for sex is important to understand their HIV burden and, ultimately, develop appropriate interventions.

### What did the researchers do and find?

- We performed meta-analyses of 87 population-based surveys from sub-Saharan Africa (2000–2020) to estimate pooled proportions of men who ever paid for sex, condom use during paid sex, HIV prevalence, and HIV prevention and treatment outcomes.

- Up to 1 in 10 sexually active men in these surveys reported ever paying for sex, and condom use during paid sex has remained suboptimal, at 68% over the last decade.

- Men living in urban areas were more likely than those in rural areas to report ever paying for sex, and younger men (15–24 years) were more likely to have paid for sex in the past 12 months.

- HIV prevalence was higher among men who ever paid for sex as compared to those who did not.

**What do these findings mean?**

- Men who pay for sex continue to constitute a distinct population subgroup at high risk of HIV acquisition and transmission. They should be recognized as a priority population for HIV prevention.

- Without increased HIV prevention efforts among clients of sex workers, including improved access to HIV testing and condom use initiatives, female sex workers and their clients will continue to be at risk of HIV acquisition.

## Introduction

Despite continued efforts to control HIV epidemics, 1.7 million new HIV infections occurred in 2019, with the greatest disease burden found in sub-Saharan Africa (SSA) [1]. In 2014, the Joint United Nations Programme on HIV/AIDS (UNAIDS) announced its objective to end AIDS by 2030 by considerably increasing diagnosis, treatment, and viral suppression among people living with HIV [2]. To achieve the HIV incidence reduction targets, interventions must prioritize key populations, which include sex workers, men who have sex with men, people who inject drugs, transgender people, and incarcerated people [3]. Key populations have unmet HIV prevention needs and contribute disproportionately to HIV transmission dynamics. Worldwide, over 60% of new adult HIV infections in 2019 were in individuals from key populations and their partners [1]. Even in high HIV prevalence settings, focusing HIV prevention approaches on key populations is important for limiting transmission [4].

Globally, sex workers experience a high HIV burden. Worldwide, an estimated 12% of female sex workers were living with HIV in 2011, reaching 37% in SSA [5]. The increased HIV acquisition risk among female sex workers is exacerbated by structural factors—including criminalization of sex work, stigma, and physical and sexual violence—which undermine sex worker engagement in HIV risk reduction behaviors and prevention [6–8]. Modeling studies suggest that the population attributable fraction of new HIV infections due to sex work ranges from less than 5% to 95%, depending on context [9–14]. This population-level impact is the result of chains of transmission linking sex workers and their clients to partners not involved in sex work [15].

Despite the central position of men who pay for sex in sexual networks, there has been comparatively little attention devoted to systematically reviewing representative epidemiological data on these men and on interventions focused on this population. Clients of sex workers are not designated, nor recognized, as a key population by UNAIDS, in part because of their lack of perceived structural vulnerabilities [1]. However, neglecting this population places the responsibility to prevent HIV transmission solely on sex workers. Developing appropriate interventions for clients of sex workers can be challenging and requires a granular understanding of the population sizes, sexual behaviors, HIV epidemiology, and uptake of HIV prevention interventions of this group. As with other key populations, clients of sex workers are hard to reach, and there can be wide variations in the definition of sex work [16]. Time–location surveys that collect information on clients of sex workers are often limited by their high non-response rates and lack of representativeness [17–19]. In contrast, nationally representative population-based surveys that collect information on paid sex may provide a promising alternative for characterizing men who pay for sex [20,21]. However, these surveys rely on self-

reports, which are susceptible to underreporting of stigmatized behaviors such as paid sex [22].

The goal of this study is to improve our understanding of the complex HIV transmission dynamics arising from sex work. To achieve this, we first synthesize national population-based surveys conducted in SSA from 2000 to 2020 that collected information on paid sex ever. Second, we use meta-analyses to estimate population sizes, lifetime number of sexual partners, condom use, and HIV prevalence, testing, and treatment outcomes among men who do, and do not, pay for sex in SSA.

## Methods

### Data sources and selection criteria

We searched for nationally representative population-based surveys conducted in SSA over the time period 2000–2020 with available microdata on ever paying for sex (Table A S1 Text). Specifically, we considered Demographic and Health Surveys (DHS), AIDS Indicator Surveys (AISs) (https://dhsprogram.com/methodology/survey-types/ais.cfm), Population-based HIV Impact Assessment (PHIA) (https://phia-data.icap.columbia.edu/), Multiple Indicator Cluster Surveys (MICS) (https://mics.unicef.org/surveys), and other country-specific population-based surveys (e.g., Kenya AIDS Indicator Survey [KAIS] and South Africa National HIV Prevalence, Incidence, Behaviour and Communication Survey [SABSSM]; Table A in S1 Text). We included all available surveys and did not exclude based on survey language.

### Variables of interest and definitions

We extracted data on paid sex (ever and past 12 months), lifetime number of sexual partners, condom use during last paid sex, HIV serostatus, HIV testing history (ever and past 12 months), antiretroviral (ARV) use (as determined by ARV biomarker data), and viral load suppression (VLS) among sexually active men. For most surveys, men were identified as having ever paid for sex if (1) they reported that any of their last 3 sex partners was a sex worker or (2) they reported either ever paying for sex or doing so in the past 12 months. Men who had never had sex were excluded.

### Data analysis

Using respondent-level data from each survey, we calculated relevant estimands, along with their 95% confidence intervals (95% CIs), for men aged 15–54 years, accounting for complex survey designs (i.e., survey weights, stratification, and clustering). We did not pool estimates if denominators were smaller than 10. We pooled outcomes using inverse-variance-weighted random-effects meta-analysis with the empirical Bayes estimator for heterogeneity. We used $I^2$ statistics to assess heterogeneity across estimates [23]. We calculated the following estimands: pooled proportions of men who paid for sex ever and in the past 12 months; pooled proportions of men who used a condom during their last paid sex; pooled proportions of men who ever tested for HIV; HIV prevalence among men who paid for sex; prevalence ratios (PRs) of HIV, HIV testing history (ever and past 12 months), ever HIV testing among people living with HIV, ARV use, and VLS among men who had paid for sex and those who had not; and mean and ratio of means (log-transformed) of lifetime number of sexual partners for men who had paid for sex and those who had not. Meta-analyses were performed on logit-transformed proportions and log-transformed PRs. Calculations were stratified by regions and by time periods (2000–2009 and 2010–2020). When calculating lifetime number of sexual partners and

the PRs for HIV and HIV testing, we standardized results by age and urban/rural residence type.

We performed univariable meta-regression to assess whether the proportion of men who paid for sex, condom use at last paid sex, HIV testing, HIV prevalence, and PRs of HIV and HIV testing (ever and in the past 12 months) varied by survey year and whether the proportion of men who paid for sex varied by age and urban/rural residence type. Meta-regression was performed using logit-transformed proportions and log-transformed PRs, and we assessed time trends by using our models to estimate outcomes in 2010 and 2020. These analyses were not pre-registered. R software was used (4.0.0), and the DHS/AIS data were extracted using the *rdhs* package [24]. Survey data were analyzed with the *survey* package [25], and meta-analyses were performed using the *metafor* package [26]. This meta-analysis was reported in accordance with MOOSE guidelines [27].

### Ethics

All analyses were performed on anonymized, de-identified data. DHS/AIS survey protocols were approved by the Institutional Review Board of ICF International in Calverton, MD, US, and other surveys (PHIA, KAIS, and SABSSM) were approved by the relevant country authorities. Ethics approval for secondary data analyses was obtained from McGill University's Faculty of Medicine Institutional Review Board (A10-E72-17B).

## Results

### Description of included surveys

Our review identified 226 nationally representative population-based surveys, of which 87 (78 DHS/AISs, 6 PHIAs, 2 SABSSMs, and 1 KAIS) included information on men ever paying for sex. These surveys were conducted in 35 countries and included 368,283 unique sexually active male respondents aged 15 to 54 years (Fig 1). Together, these 35 countries represent 95% of men in SSA [28]. Survey questions were sufficiently standardized for them to be pooled (Table B in S1 Text). Twenty-six countries had more than 1 included survey, and the median year of data collection was 2012. Under two-thirds of surveys had information on HIV sero-prevalence (number of surveys [$N_s$] = 52), but only 9% had information on ARV biomarkers ($N_s$ = 8), and 10% on VLS ($N_s$ = 9).

All 87 surveys were included to calculate the proportion of men who reported ever paying for sex (Fig 2). Fifty-two surveys were included in analyses of HIV prevalence and PRs (Fig 2); the Zambia 2013–14 DHS survey was not included in this analysis because of concerns about the accuracy of the results of the HIV testing algorithm assay [29]. Surveys were included in other analyses based on inclusion of relevant questions.

### Population size and lifetime number of sexual partners

The pooled proportion of sexually active men who reported ever paying for sex in SSA was 8.0% (95% CI 6.1%–10.3%, $N_s$ = 87, $I^2$ = 100%; Fig 3; Table C in S1 Text), with the highest proportions in Central and Eastern Africa, at 11.9% and 11.3%, respectively. Proportions were similar for surveys conducted from 2010 onwards (8.5%, 95% CI 6.4%–11.2%, $N_s$ = 64, $I^2$ = 100%) and before 2010 (6.7%, 95% CI 4.0%–10.9%, $N_s$ = 23, $I^2$ = 100%; Table D in S1 Text). There were no time trends in the proportion of men who paid for sex from 2000 to 2020 (Table E in S1 Text). Men residing in urban areas were more likely to report ever paying for sex (9.7%, 95% CI 7.3%–12.7%) than those from rural areas (7.1%, 95% CI 5.2%–9.6%; Table F in S1 Text). The pooled proportion of sexually active men who paid for sex in the past 12

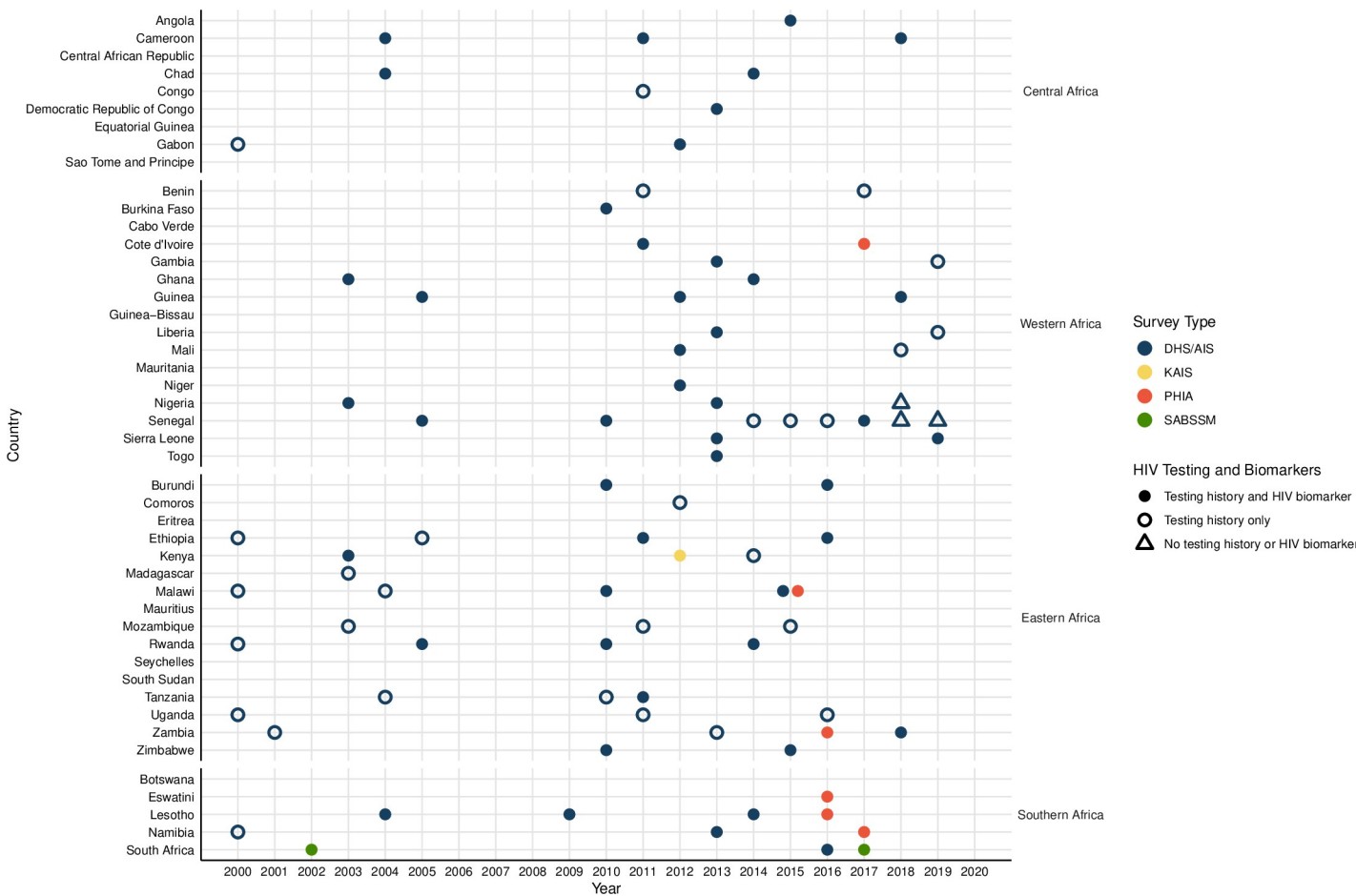

**Fig 1. Surveys with questions about ever paying for sex, by country and year, 2000–2020.** Points represent population-based surveys conducted in sub-Saharan Africa from 2000 to 2020 and asking men about ever paying for sex. Circles represent surveys with data on HIV testing, while triangles represent surveys without data on HIV testing. Filled in points represent surveys that include HIV biomarker testing, while empty points represent surveys that did not have biomarker testing. AIS, AIDS Indicator Survey; DHS, Demographic and Health Surveys; KAIS, Kenya AIDS Indicator Survey; PHIA, Population-based HIV Impact Assessment; SABSSM, South African National HIV Prevalence, Incidence, Behaviour and Communication Survey.

months was 2.7% (95% CI 2.1%–3.5%, $N_s$ = 79, $I^2$ = 100%). Younger men (15–24 years) were more likely to report paying for sex in the past 12 months (5.1%, 95% CI 3.6%–7.1%) than those aged 35–54 years (2.2%, 95% CI 1.5%–3.2%; Table G in S1 Text).

Men who paid for sex had an average 12.0 lifetime sexual partners (standardized; 95% CI 10.9–13.1, $N_s$ = 68, $I^2$ = 100%). The average lifetime number of partners was highest in Central Africa (19.6, 95% CI 15.5–24.8, $N_s$ = 9, $I^2$ = 100%) and lowest in Eastern Africa (10.2, 95% CI 9.1–11.3, $N_s$ = 24, $I^2$ = 100%). Across all 4 regions, men who paid for sex had consistently more lifetime sexual partners, with an average 2.3 times more partners compared to men who did not pay for sex (standardized; 95% CI 2.1–2.4, $N_s$ = 68, $I^2$ = 100%; Fig C in S1 Text).

## Condom use

Among men who reported paid sex in the past year, 62.2% used a condom the last time they paid for sex (95% CI 57.4%–66.7%, $N_s$ = 84, $I^2$ = 97%; Fig 4; Fig D in S1 Text), with the highest proportion in Southern Africa, at 76.6% (95% CI 65.8%–84.7%, $N_s$ = 11, $I^2$ = 88%), and lowest proportion in Eastern Africa, at 55.2% (95% CI 46.5%–63.6%, $N_s$ = 35, $I^2$ = 98%). We did not

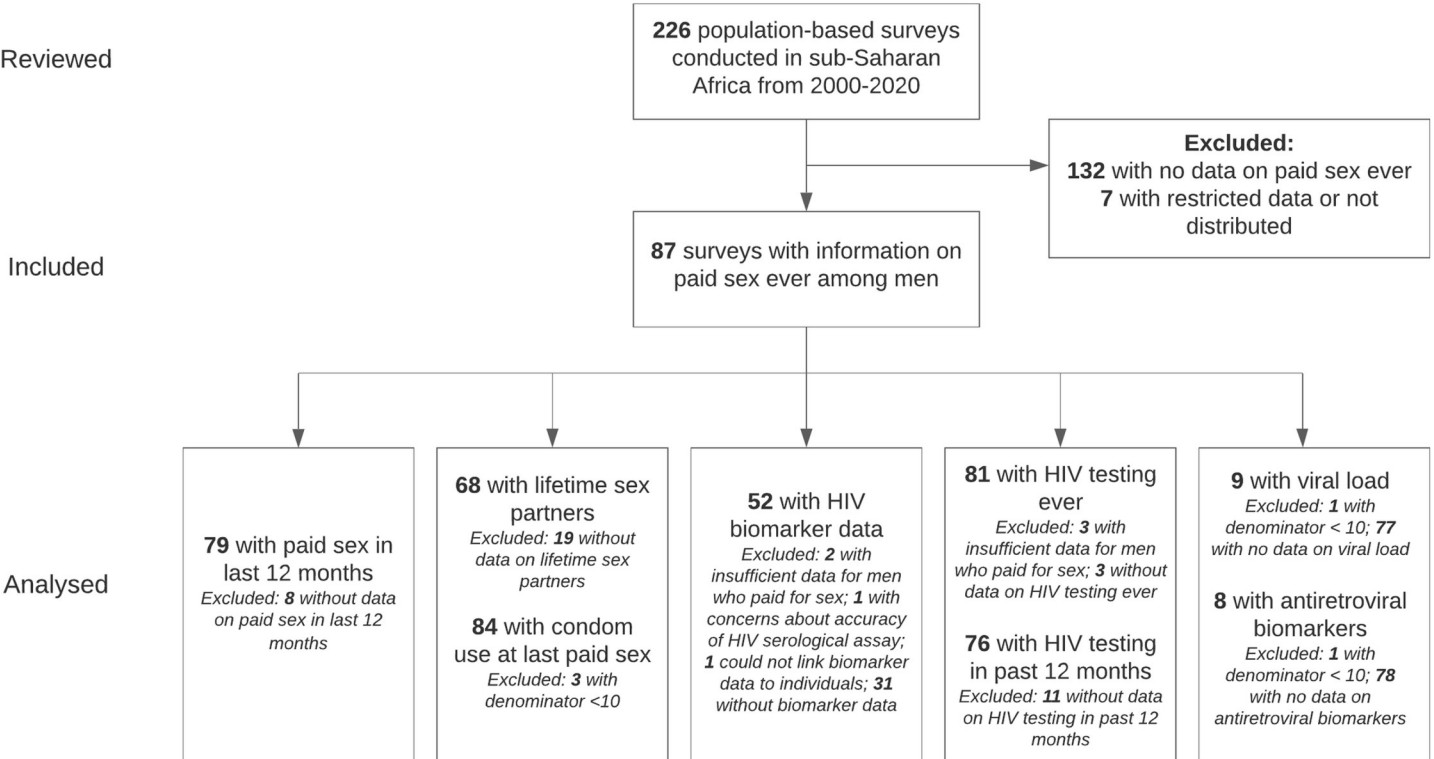

**Fig 2. Flowchart of population-based survey inclusion in each analysis.** A total of 226 population-based surveys conducted in sub-Saharan Africa from 2000 to 2020 were reviewed, and 87 were identified as having information on paid sex ever among men. Surveys were included in each analysis based on availability of relevant information.

compare condom use at last paid sex with condom use at last sex among men who did not pay for sex because condom use depends strongly on partner type. Condom use at last paid sex increased over time (odds ratio per decade 1.07, 95% CI 1.04–1.11; estimate for 2010: 60.4%, 95% CI 55.8%–64.7%; estimate for 2020: 75.7%, 95% CI 70.6%–80.2%; Table E in S1 Text). Pooling by time period, condom use at last paid sex was higher for surveys conducted from 2010 onwards (67.5%, 95% CI 63.9%–70.9%, $N_s$ = 61) than for surveys conducted before 2010 (46.6%, 95% CI 37.9%–55.4%, $N_s$ = 23; Table D and Fig E in S1 Text).

## HIV prevalence

The pooled HIV prevalence among men who paid for sex was 5.1% (95% CI 3.4%–7.5%, $N_s$ = 52, $I^2$ = 98%; Fig F in S1 Text). This varied greatly across regions and was highest in Southern Africa and lowest in Central Africa. The pooled, standardized PR between HIV among men who paid for sex and those who did not was 1.50 (95% CI 1.31–1.72, $N_s$ = 52, $I^2$ = 87%; Fig 5). Men who paid for sex were more likely to be living with HIV than men who did not pay for sex in all 4 regions, with the highest ratio in Western Africa (1.67, 95% CI 1.10–2.53, $N_s$ = 18, $I^2$ = 63%).

HIV prevalence among men who paid for sex decreased slightly over time (odds ratio per year 0.98, 95% CI 0.95–1.01; estimate for 2010: 5.5%, 95% CI 3.7%–8.2%; estimate for 2020: 3.6%, 95% CI 1.6%–8.1%), but uncertainty was large, and this finding was also consistent with a stable HIV prevalence (Table E in S1 Text). Pooled HIV prevalence was lower for surveys conducted from 2010 onwards compared to surveys conducted before 2010 (Table D in

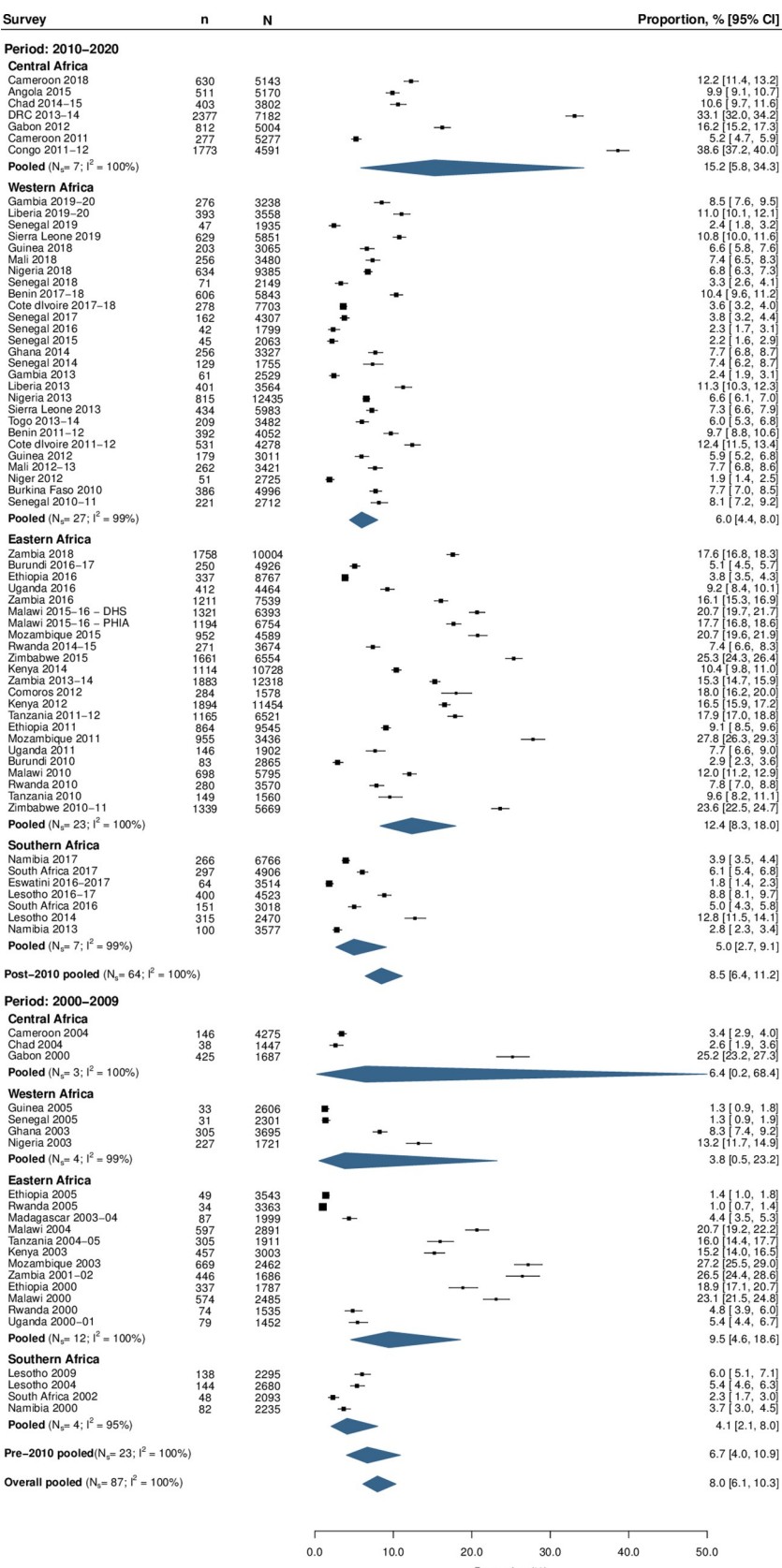

**Fig 3. Forest plots of proportions of sexually active men who ever paid for sex.** Data from 87 population-based surveys (78 DHS/AISs, 6 PHIAs, 2 SABSSMs, 1 KAIS) were collected and meta-analyses conducted to determine the proportion of men who have ever paid for sex. Pooled proportions were calculated for each region for both post-2010 (2010–2020) and pre-2010 (2000–2009) surveys, and overall. AIS, AIDS Indicator Survey; DHS, Demographic and Health Surveys; DRC, Democratic Republic of the Congo; KAIS, Kenya AIDS Indicator Survey; PHIA, Population-based HIV Impact Assessment; SABSSM, South African National HIV Prevalence, Incidence, Behaviour and Communication Survey.

S1 Text). There was a small reduction with time in the HIV PR between men who paid for sex and those who did not (Table E in S1 Text).

### HIV testing

Men who paid for sex were more likely to report having tested for HIV ever and in the past 12 months (Figs G and H in S1 Text) compared to those who did not pay for sex. The pooled, standardized PR for having ever tested for HIV was 1.14 (95% CI 1.06–1.24, $N_s = 81$, $I^2 = 98\%$), while the pooled, standardized PR for testing in the past 12 months was 1.09 (95% CI 1.00–1.18, $N_s = 76$, $I^2 = 95\%$). There were no time trends in the PR of HIV testing (Table E in S1 Text).

Across regions, men who paid for sex were generally more likely to have been tested for HIV compared to men who did not pay for sex. The PR of ever HIV testing was highest in Western Africa and lowest in Central Africa (Fig G in S1 Text), while the PR of testing in the past 12 months was highest in Central Africa and lowest in Eastern Africa (Fig H in S1 Text).

Among men who paid for sex, the pooled proportion ever tested for HIV was 34.4% (95% CI 26.9%–42.7%, $N_s = 81$, $I^2 = 99\%$; Table C in S1 Text). This proportion was highest in Southern Africa and lowest in Western Africa. Lifetime HIV testing among men who paid for sex increased over time (odds ratio per year 1.15, 95% CI 1.10–1.20; estimate for 2010: 31.9%, 95% CI 25.5%–39.1%; estimate for 2020: 64.9%, 95% CI 52.0%–75.9%; Table E in S1 Text).

Among those living with HIV, the proportion ever tested for HIV was similar among men who paid for sex and those who did not (PR 0.96, 95% CI 0.88–1.05, $N_s = 18$, $I^2 = 83\%$; Fig I in S1 Text). When pooled by region, men living with HIV who paid for sex were less likely to have ever tested for HIV than men living with HIV who did not pay for sex in all 4 regions, although all confidence intervals crossed the null (Fig I in S1 Text).

### ARV use and VLS

Few surveys with sufficient denominators included information on ARV biomarkers ($N_s = 8$) and viral load ($N_s = 9$). For people living with HIV, there was no evidence of a difference in ARV biomarker coverage among men living with HIV who paid for sex compared to those who did not (PR = 1.01, 95% CI 0.86–1.18, $N_s = 8$, $I^2 = 71\%$; Fig J in S1 Text). Results were similar for VLS (PR = 1.00, 95% CI 0.86–1.17, $N_s = 9$, $I^2 = 51\%$; Fig K in S1 Text).

### Discussion

In this study, we systematically analyzed 87 population-based surveys conducted over 2 decades in SSA. From 2000 to 2020, 8% of sexually active men in SSA reported ever paying for sex, and this proportion was higher in urban areas. Men who paid for sex were 50% more likely to be living with HIV compared to men who did not pay for sex, and only 68% of men in the last decade reported using a condom during their last paid sex. Men who paid for sex had a slightly higher probability of having ever tested for HIV, but ARV use and VLS among men

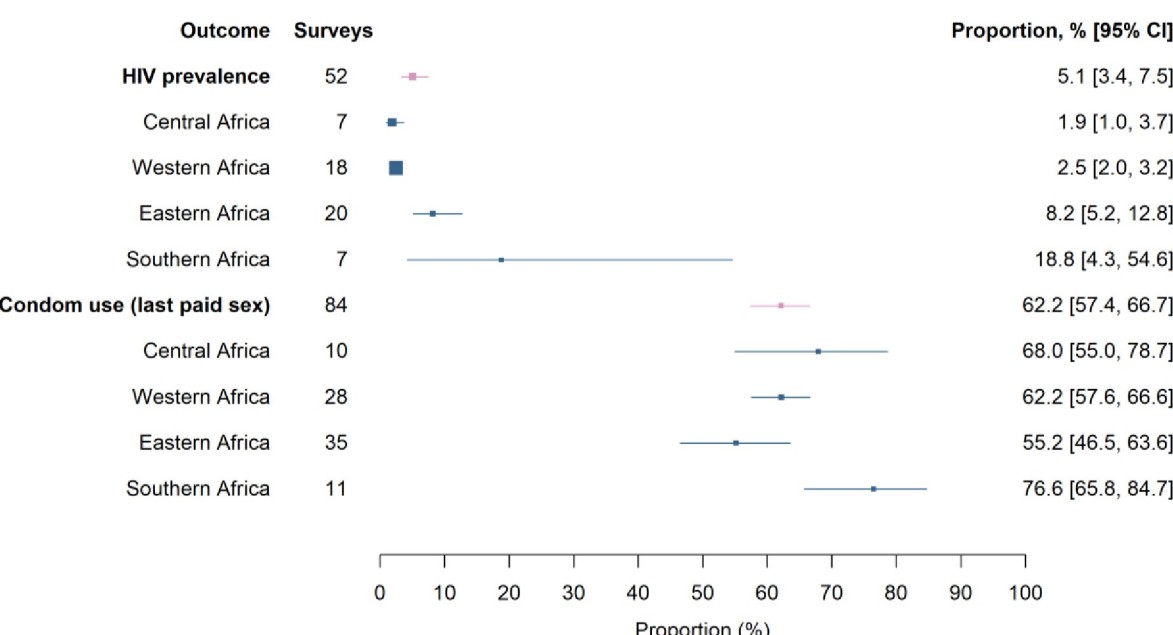

| Outcome | Surveys | Proportion, % [95% CI] |
|---|---|---|
| **HIV prevalence** | 52 | 5.1 [3.4, 7.5] |
| Central Africa | 7 | 1.9 [1.0, 3.7] |
| Western Africa | 18 | 2.5 [2.0, 3.2] |
| Eastern Africa | 20 | 8.2 [5.2, 12.8] |
| Southern Africa | 7 | 18.8 [4.3, 54.6] |
| **Condom use (last paid sex)** | 84 | 62.2 [57.4, 66.7] |
| Central Africa | 10 | 68.0 [55.0, 78.7] |
| Western Africa | 28 | 62.2 [57.6, 66.6] |
| Eastern Africa | 35 | 55.2 [46.5, 63.6] |
| Southern Africa | 11 | 76.6 [65.8, 84.7] |

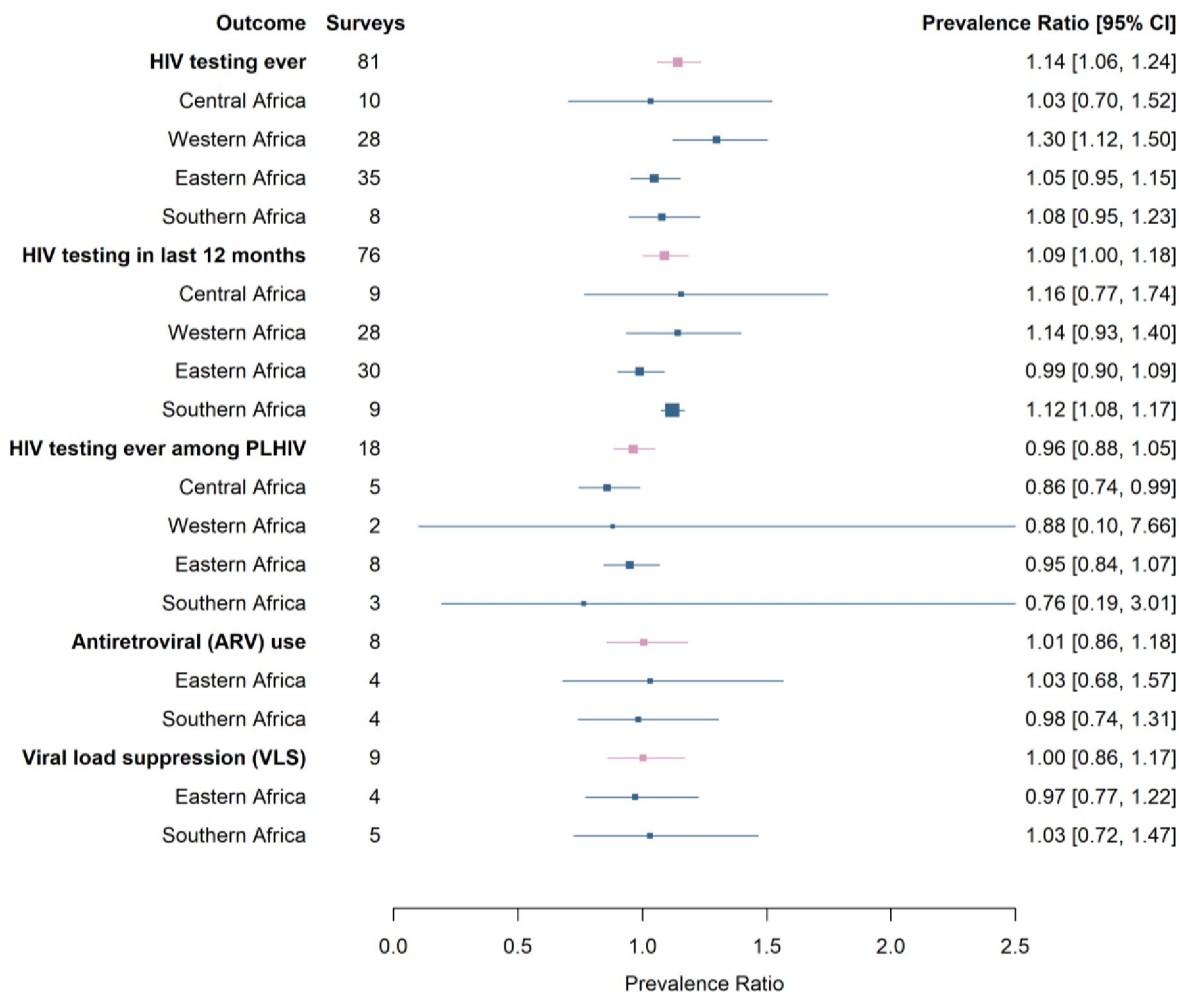

| Outcome | Surveys | Prevalence Ratio [95% CI] |
|---|---|---|
| **HIV testing ever** | 81 | 1.14 [1.06, 1.24] |
| Central Africa | 10 | 1.03 [0.70, 1.52] |
| Western Africa | 28 | 1.30 [1.12, 1.50] |
| Eastern Africa | 35 | 1.05 [0.95, 1.15] |
| Southern Africa | 8 | 1.08 [0.95, 1.23] |
| **HIV testing in last 12 months** | 76 | 1.09 [1.00, 1.18] |
| Central Africa | 9 | 1.16 [0.77, 1.74] |
| Western Africa | 28 | 1.14 [0.93, 1.40] |
| Eastern Africa | 30 | 0.99 [0.90, 1.09] |
| Southern Africa | 9 | 1.12 [1.08, 1.17] |
| **HIV testing ever among PLHIV** | 18 | 0.96 [0.88, 1.05] |
| Central Africa | 5 | 0.86 [0.74, 0.99] |
| Western Africa | 2 | 0.88 [0.10, 7.66] |
| Eastern Africa | 8 | 0.95 [0.84, 1.07] |
| Southern Africa | 3 | 0.76 [0.19, 3.01] |
| **Antiretroviral (ARV) use** | 8 | 1.01 [0.86, 1.18] |
| Eastern Africa | 4 | 1.03 [0.68, 1.57] |
| Southern Africa | 4 | 0.98 [0.74, 1.31] |
| **Viral load suppression (VLS)** | 9 | 1.00 [0.86, 1.17] |
| Eastern Africa | 4 | 0.97 [0.77, 1.22] |
| Southern Africa | 5 | 1.03 [0.72, 1.47] |

**Fig 4. Pooled estimates of HIV prevalence, condom use at last paid sex, HIV testing history, antiretroviral use, and viral load suppression among men who paid for sex, overall and stratified by sub-Saharan Africa regions.** Meta-analysis was performed for each outcome, and pooled proportions were calculated by region and overall. Analyses for HIV prevalence and HIV testing history were standardized by age and urban/rural residence type. PLHIV, people living with HIV.

who paid for sex were similar to those among men who did not pay for sex, although the evidence for ARV use and VLS were limited to fewer countries.

We found important regional variations in reports of paid sex. A higher proportion of men reported ever paying for sex in Central and Eastern Africa compared to Western and Southern Africa, which is consistent with a 2006 systematic review [16]. The overall proportion of sexually active men who paid for sex in the past 12 months was 2.7%, which is lower than an estimate of 4.3% from a review of 2010–2016 DHS surveys [30]. Differences can be explained by the present study including more surveys, representing more participants and 8 more countries. Also, our main analyses use lifetime measures of paid sex, which may be less prone to underreporting and social desirability bias than measures of recent paid sex. However, our population size estimates for clients of sex workers are probably lower bounds because of potential non-disclosure of paid sex. For example, the proportion of adult women engaged in sex work is estimated to range from 0.4% to 4.3% [31]. Given these numbers, it is unlikely that our estimate of 8% of men paying for sex would be sufficient to sustain this number of women engaging in sex work, although a study from Rwanda reported similar numbers of female sex workers and clients [32]. All surveys included here used face-to-face interviewer-administered questionnaires. Responses could be affected by social desirability bias, and this bias could be greater than for alternative confidential survey methods such as polling booth surveys [22].

Consistent with regional variations in population HIV prevalence, HIV prevalence among men who paid for sex was highest in Southern Africa and lowest in Central and Western Africa [1]. Men who paid for sex were more likely to be living with HIV than men who did not pay for sex, which is consistent with a 2008 analysis [33]. A similar finding was also found for female sex workers, whose odds of living with HIV were 12 times higher than that of all women aged 15–49 years [5]. HIV prevalence among men who paid for sex was lower in surveys conducted from 2010 onwards compared to surveys conducted before 2010, but uncertainty was large, and we cannot rule out that prevalence among this group remained stable. Although HIV prevalence was lowest in Western Africa, this region had the highest HIV PR comparing men who pay for sex with those who do not. In these settings, adding interventions that focus on the unmet prevention needs of men who pay for sex may be more cost-effective than those focused on the general population [34].

Men who paid for sex were more likely to have ever tested for HIV across regions and time. The PR for lifetime HIV testing between men who paid for sex and men who did not was highest in Western Africa and lowest in Central Africa. Higher risk perception encouraging testing, or greater availability of testing in areas with higher HIV burden, may explain this result [35,36]. Men living with HIV who paid for sex were equally likely to have been tested as those who did not paid for sex, which could have implications for knowledge of HIV status in this group. A recent study found that diagnosis coverage and time to diagnosis in SSA have drastically improved over the last decade, but in 2020 the largest group of individuals unaware of their status was men [37]. Distribution of HIV self-tests to sex workers, who can then distribute the tests to peers, clients, and partners, may further improve knowledge of status among men who pay for sex [38]. This approach is preferable to interventions such as index testing that require sex workers to disclose the identity of their male clients, which could put the sex workers at risk of violence, loss of sex work income, or both [6].

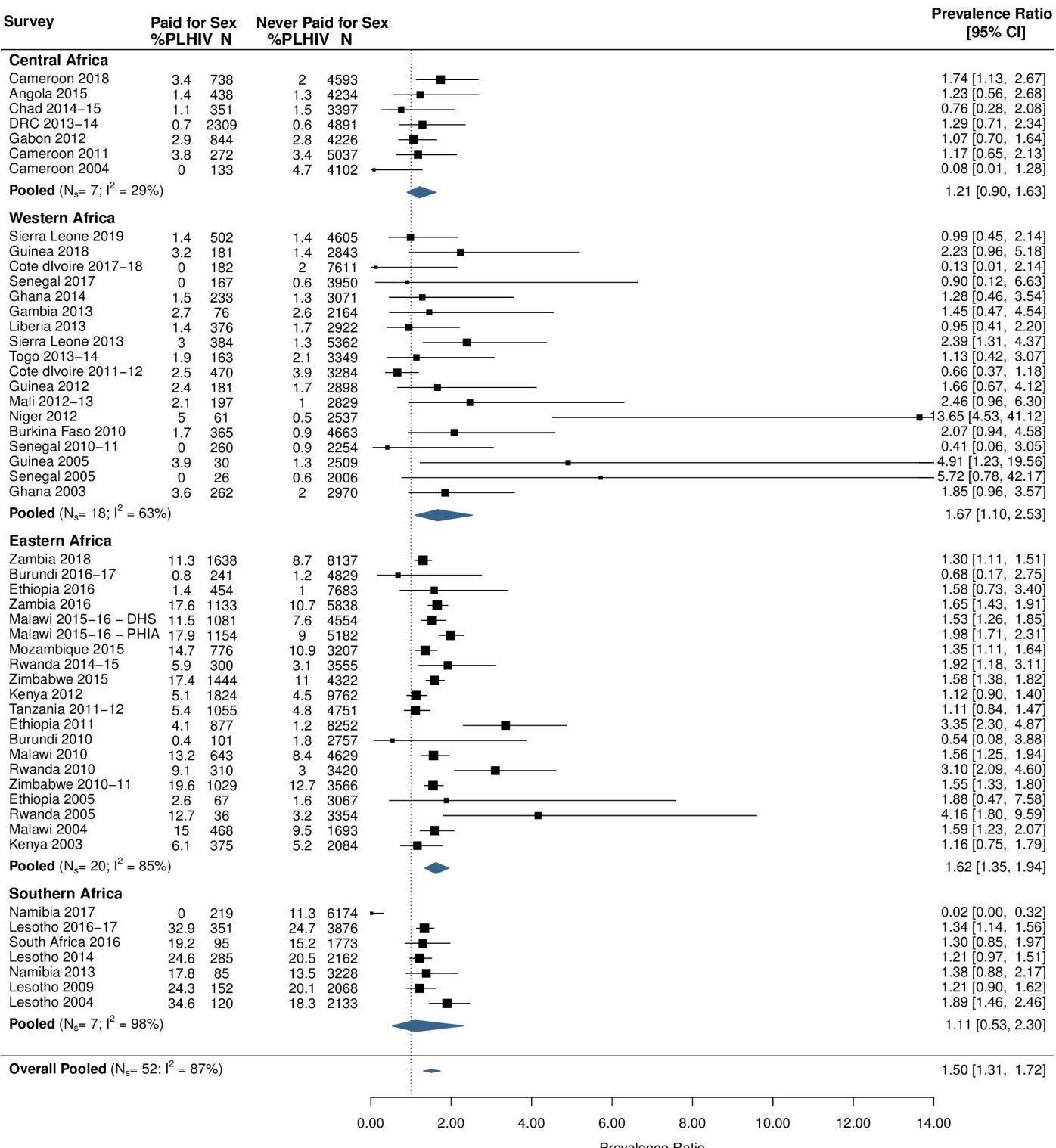

**Fig 5. Forest plot of standardized HIV prevalence ratios for men who have ever paid for sex compared to men who have never paid for sex.** HIV biomarker data from 52 population-based surveys were collected and meta-analyses conducted to determine HIV prevalence ratios for men who have ever paid for sex compared to men who have not. Prevalence ratios are standardized by age and urban/rural residence type. Pooled prevalence ratios were calculated for each region and overall. DHS, Demographic and Health Surveys; DRC, Democratic Republic of the Congo; PHIA, Population-based HIV Impact Assessment; PLHIV, people living with HIV.

As men who pay for sex often also have female partners not involved in sex work, they may disproportionally contribute to population-level HIV transmission if virally unsuppressed [9,15,39]. Our results suggest comparable ARV use and VLS levels among men who paid for sex and men who did not. However, these estimates are based on a small number of surveys from 2012–2017, highlighting important data gaps. For SSA as a whole, the 2020 estimates of ARV use and VLS remain below UNAIDS targets, and men may be less likely to initiate and adhere to ARV treatment than women [1,40,41]. Treatment access can be facilitated by services targeted to men who are more likely to frequent sex workers, such as migrant laborers, long-distance truck drivers, mine workers, and other men who travel for work [42]. Improving access to treatment for men who pay for sex is especially important as, from 2010 onwards, only 68% of men used a condom the last time they paid for sex. A recent analysis of 29 DHS surveys from 2010–2019 found that, among men who reported condom use at their last paid sex, 84% reported consistent condom use during paid sex [43], which is higher than our estimates. Since the survey instruments only asked about consistent condom use if men reported condom use at last paid sex, we would expect consistent condom use measures to be higher. Altogether, these results suggest that, when men pay for sex and use condoms, they tend to do so consistently. Nevertheless, clients of sex workers often have decisive power over condom use during paid sex, and global evidence suggests higher HIV prevalence among clients of sex workers who do not use condoms [9,44,45]. For these reasons, continued condom use promotion in this population is strongly warranted.

Our results should be interpreted considering several limitations. First, population-based surveys of sexual behaviors depend on self-reports, so estimates could be affected by recall and social desirability biases [46–48]. Use of a lifetime measure of paid sex may have alleviated underreporting, but measures could still be underestimated, and, in the case of measures of association, this underestimation could attenuate the effect sizes towards the null. Confidential measures like polling booth surveys or audio computer-assisted self-interviews may improve accuracy [22,49,50]. Second, survey instruments captured men who have "paid" or, in a few instances, "given money, gifts, or favors in exchange" for sex. As money can be exchanged for sex outside of sex work, we cannot be certain that all men in our population are clients of sex workers. There are many sex work typologies, and transactional sex that involves exchanging gifts or favors may not have been reported as paid sex. For instance, relationships between male "sugar daddies" and younger women may be an important type of transactional sex that is probably not entirely captured in our surveys [51]. This could partly explain the smaller population size estimates for the Southern Africa region. For example, 18% of men in 2 South Africa provinces reported "ever having sex with a woman in prostitution," but 66% reported having had some type of transactional sex [52]. Third, most surveys do not specify the paid partner's gender when asking about paid sex, and we cannot be certain that all men in our analytical sample have paid for sex with a woman. However, the proportion of men who have sex with men in this region is estimated to be small [53]. Fourth, the included surveys had slightly different questionnaires and sampling strategies. Nevertheless, questions were largely similar, and using these multiple data sources allowed us to integrate information from more countries and respondents. Finally, few surveys had information on ARV treatment and VLS, so these estimates may not be generalizable to all regions.

Strengths of this study include our exhaustive analysis of all available population-based surveys with information on men who ever paid for sex in SSA, without restriction to any survey type. We synthesized new information on the epidemiology of HIV and the HIV prevention and treatment cascades among men who pay for sex. Our large sample size allowed investigation of patterns by regions and over time, and we estimated adjusted PRs using standardization to control for the effects of age and urban or rural area of residence.

## Conclusion

Up to 1 in 10 men report ever paying for sex in SSA. To more accurately determine population sizes of men who pay for sex, improved confidential methods should be employed. Compared to those who have never paid for sex, men who have paid for sex have over double the lifetime number of partners, are more likely to be living with HIV, and, despite higher testing, could be less likely to know their status in some regions. Condom use initiatives and improved access to HIV testing campaigns are required to prevent HIV transmission from clients to sex workers and to their other sexual partners. These results suggest that men who pay for sex continue to constitute a distinct population subgroup at high risk of HIV acquisition and transmission, and that they should be recognized as a priority population for HIV prevention.

## Supporting information

**S1 Text. Additional information on the surveys, detailed forest plots, and complementary statistical analyses.** Table A: List of surveys considered and justifications for exclusion. Table B: Characteristics of population-based surveys conducted between 2000 and 2020 with available microdata included in analyses. Table C: Number of surveys, pooled estimates, confidence intervals, prediction intervals, and $I^2$ values by region and overall for each outcome. Table D: Pooled estimates, confidence intervals, prediction intervals, and $I^2$ for 2000–2009 and 2010–2020 for prevalence of paying for sex, condom use at last paid sex, and HIV prevalence and testing among men who have paid for sex. Table E: Results of univariate meta-regression for survey year. Table F: Pooled estimates, confidence intervals, prediction intervals, and $I^2$ values for prevalence of paying for sex ever and in the past 12 months by urban/rural residence type. Table G: Pooled estimates, confidence intervals, prediction intervals, and $I^2$ values for prevalence of paying for sex ever and in the past 12 months by age groups. Fig A: Flow charts of "HIV testing history" and "men who have ever paid for sex." Fig B: Men ever paying for sex over time, by country. The proportion of sexually active men reporting ever paying for sex was calculated for 87 population-based surveys and plotted over time for countries with 3 or more surveys. Fig C: Bar graph of standardized mean lifetime number of sex partners for men who have paid for sex compared to men who have not, by survey. Fig D: Forest plot of proportion of men who paid for sex who reported condom use at last paid sex. Fig E: Condom use at last paid sex over time, by country. Fig F: Forest plot of standardized HIV prevalence for men who have paid for sex. Data from 52 population-based surveys was collected and meta-analysis conducted to determine HIV prevalence among men who reported having paid for sex. Prevalence is standardized by age and urban/ rural residence type. Proportions were pooled by region and overall. Fig G: Forest plot of standardized prevalence ratios for HIV testing ever among men who have paid for sex compared to men who have not. Fig H: Forest plot of standardized prevalence ratios for HIV testing in the last 12 months among men who have paid for sex compared to men who have not. Fig I: Forest plot of standardized prevalence ratios for HIV testing ever among men living with HIV who have paid for sex compared to men who have not. Fig J: Forest plot of prevalence ratios of antiretroviral use among men living with HIV who have paid for sex compared to men who have not. Fig K: Forest plot of prevalence ratios of viral load suppression among men living with HIV who have paid for sex compared to men who have not.
(DOCX)

## Acknowledgments

We thank Dr. Michel Alary for useful feedback and suggestions on this study.

## Author Contributions

**Conceptualization:** Lycias Zembe, Jeffrey W. Eaton, Marie-Claude Boily, Mathieu Maheu-Giroux.

**Data curation:** Caroline Hodgins.

**Formal analysis:** Caroline Hodgins, James Stannah, Salome Kuchukhidze.

**Funding acquisition:** Mathieu Maheu-Giroux.

**Supervision:** Lycias Zembe, Jeffrey W. Eaton, Marie-Claude Boily, Mathieu Maheu-Giroux.

**Writing – original draft:** Caroline Hodgins, James Stannah, Salome Kuchukhidze, Mathieu Maheu-Giroux.

**Writing – review & editing:** Caroline Hodgins, James Stannah, Salome Kuchukhidze, Lycias Zembe, Jeffrey W. Eaton, Marie-Claude Boily, Mathieu Maheu-Giroux.

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
