## [Editor Report · Decision Letter 0]

15 Jun 2021

Dear Dr Maheu-Giroux, 

Thank you for submitting your manuscript entitled "HIV prevalence, population sizes, and HIV prevention among men who paid for sex in sub-Saharan Africa: a meta-analysis of 82 population-based surveys (2000-2020)" for consideration by PLOS Medicine.

Your manuscript has now been evaluated by the PLOS Medicine editorial staff and I am writing to let you know that we would like to send your submission out for external peer review.

Please re-submit your manuscript within two working days, i.e. by Jun 17 2021 11:59PM.

Kind regards,

Beryne Odeny

Associate Editor

PLOS Medicine

---

## [Decision Letter · Decision Letter 1]

14 Aug 2021

Dear Dr. Maheu-Giroux,

Thank you very much for submitting your manuscript "HIV prevalence, population sizes, and HIV prevention among men who paid for sex in sub-Saharan Africa: a meta-analysis of 82 population-based surveys (2000-2020)" (PMEDICINE-D-21-02581R1) for consideration at PLOS Medicine. 

Your paper was discussed among the editors and sent to independent reviewers, including a statistical reviewer. The reviews are appended at the bottom of this email and any accompanying reviewer attachments can be seen via the link below:

[LINK]

In light of these reviews, we will not be able to accept the manuscript for publication in the journal in its current form, but we would like to invite you to submit a revised version that addresses the reviewers' and editors' comments fully. You will appreciate that we cannot make a decision about publication until we have seen the revised manuscript and your response, and we expect to seek re-review by one or more of the reviewers. 

We hope to receive your revised manuscript by Sep 06 2021 11:59PM. Please email us (plosmedicine@plos.org) if you have any questions or concerns.

Please let me know if you have any questions, and we look forward to receiving your revised manuscript. 

Sincerely,

Richard Turner PhD, for Beryne Odeny

Senior editor, PLOS Medicine

rturner@plos.org

In the title, please move the dates before the colon. 

Please combine the "Methods" and "Findings" subsections of your abstract. 

Please add a new final sentence to the combined subsection, which should begin "Study limitations include ..." or similar and should quote 2-3 of the study's main limitations. 

Please remove the information on funding from the abstract.

After the abstract, we will need to ask you to add a new and accessible "Author summary" section in non-identical prose. You may find it helpful to consult one or two recent research papers published in PLOS Medicine to get a sense of the preferred style. 

Early in the Methods section, please state whether the analysis was registered and whether or not it had a protocol or prespecified analysis plan, and if so attach the document(s) as a supplementary file(s), referred to in the text. 

Throughout the text, please style reference call-outs as follows: "... may explain this [30,31]." (noting the absence of spaces within the square brackets). 

Please remove the information on funding and competing interests from the end of the main text. In the event of publication, this will appear in the article metadata, via entries in the submission form. 

In the reference list, please convert italics and boldface to plain text.

Noting reference 8 and others, please list 6 author names rather than 3, followed where appropriate with "et al.".

Noting reference 32, please add "[preprint]" to all preprints cited. 

Please use the journal name abbreviation "PLoS ONE" in the reference list.

Please complete a checklist for the most appropriate reporting guideline, e.g., PRISMA (we suggest PRISMA 2020, https://journals.plos.org/plosmedicine/article?id=10.1371/journal.pmed.1003583), and attach this as a supplementary file, labelled "S1_PRISMA_Checklist" or similar and referred to as such in your Methods section. 

In the checklist, please refer to individual items by section (e.g., "Methods") and paragraph number, not by line or page numbers as these generally change in the event of publication. 

Comments from the reviewers:

*** Reviewer #1: 

[See attachment]

Michael Dewey

*** Reviewer #2: 

This is an excellent study that is adding to the small number of evidence on the clients of sex workers.

However, the authors did not address many of the issues they have highlighted. Using your critical minds and wealth of knowledge, what recommendations/ suggestions to you have for the public health authorities, the FSWs or to even clients of sex workers?

It would have been interesting to know:

-their policy recommendations for public health authorities?

- any recommendations for public health authorities using sex workers are index clients?

-what prevention approaches to tackle the high HIV prevalence in Western Africa versus the Southern?

-What services should be added in the health centers to reach out to clients of sex workers?

As a public health expert, this study is equality interesting to me as an evidence paper and as a study to guide policy recommendations. Unfortunately, the latter is missing. 

*** Reviewer #3: 

The study is very useful especially now that HIV prevalence remain stable in some countries in Sub Saharan Africa (SSA). It is appreciated to see that the authors have focused on a specific population group (Men) which provides more specific information regarding their vulnerability to HIV infections through pay for sex. 

However, there are a few aspects that need to be incorporated:

- it is important to clarify starting with the abstract on which countries in SSA Africa the data was analyzed

- It is not clear on whether men who were involved in the study were heterosexual or not, I think the sexual orientation need to be stipulated clearly

- It is also important that the authors clarify in the abstract section about whether men paid for sex with women or with men? this is not clearly stated in the abstract

- An implication of the study on access to treatment will be useful 

- Some additional lines that would contribute to the existing paradox that women also pay for sex (Mtenga et al. AIDS Res Ther (2018) 15:12 https://doi.org/10.1186/s12981-018-0199-6), will make a an important argument. 

*** Reviewer #4: 

I would like to appreciate the authors for the great work. Conceptually, I found the paper very coherent and structured in a well thought out fashion. With pleasure, I would like to provide a few suggestions on the manuscript.

First, motivation of the authors to do the paper has been mentioned as "less attention" drawn to interventions aimed at targeting clients of sex workers. However, this justification could not tell the convincing reason for doing the study as the study focuses on HIV epidemiology, including other related issues, among these groups. If you find that interventions on men paying for sex are given less attention, then the study should have been on finding reason for "why this happens". But now, your study is on HIV burden, HIV testing, condom use etc, and the driver for carrying out this should have been explained differently like for instance, limited knowledge base on this area. Simply put, linking a study focused on HIV burden with a justification using a statement like inattention to prevention [of HIV among those groups] did not work here. Interventions might not be given attention but as the same time HIV burden can be known. So, this leads us to that, there might be many other important factors that potentially prevent policy makers and intervention designers from drawing attention to interventions, and lack of knowledge on HIV epidemiology [the other issues this study has addressed] may still NOT be one of the factors. 

So, I suggest that it would be highly useful to succinctly explicate the direct contribution of the paper towards reducing men running into the paid sex practice as well as to our knowledge base on this area.

Second, the selection process of the papers lacks some clarity. How did the authors restrict surveys between 2010-2020, for instance. It is also good to use PRIMSA adopted to the IPD, Individual Participant Data meta-analysis. 

Third, the chosen of the random effect model was not justified; why not fixed effect model rather. Give reason. I found the I-squared statistics to be extremely high, suggesting considerable between-study variation in terms of the variables you studied. Given this variability, do you think producing a single overall estimate through pooling is a good practice, as the papers are too different already. Meta-analysists often try fitting meta-regression to explain the variation; in this study, meta-regression could have been done better by including more confounding variables in the model to get unbiased results. Also, under variables, make it clear whether all of these are outcome variables, and mention your independent variables, if any.

Fourth, of the three design elements of a complex design study like DHS, the authors take into account only clustering and unequal probability of selection, via, weighting. However, stratification is missing, and failing to take this problem into account would result in estimation of standard errors that are biased. DHS experts highly recommend researchers to make account of all the three design elements of complex designs like DHS. I would see as a big limitation of the paper as your confidence intervals are likely to be biased. 

Finally, these days, countries are moving towards ensuring health equality within their population. International agreements like SDG calls for equity; that means, while the time is to look at health outcomes between different population groups within a country, your study was about aggregating country-level information into a regional level. How do you see the implication of the paper for the wellbeing of different segments of population in each individual country in light of SDG? 

Within country inequality analysis of HIV prevalence among men paying for sex is very important to understand who these group of men are and why they are being engaged in the activity. Because, all men in this high-risk behavior are not homogenous population group and how their engagement in this risky sexual behavior is affected by contexts they live in remains a huge research question that could substantially contribute towards the SDG related with HIV. To the contrary, you did aggregate based analysis. Can explain more on this. 

Regards, 

Gebretsadik Shibre

***

[LINK]

---

## [Decision Letter · Decision Letter 2]

12 Oct 2021

Dear Dr. Maheu-Giroux,

Thank you very much for re-submitting your manuscript "HIV prevalence, population sizes, and HIV prevention among men who paid for sex in sub-Saharan Africa (2000-2020): a meta-analysis of 87 population-based surveys" (PMEDICINE-D-21-02581R2) for review by PLOS Medicine.

I have discussed the paper with my colleagues and the academic editor and it was also seen again by one reviewer. I am pleased to say that provided the remaining editorial and production issues are dealt with we are planning to accept the paper for publication in the journal.

[LINK]

We look forward to receiving the revised manuscript by Oct 19 2021 11:59PM.   

Sincerely,

Beryne Odeny, 

PLOS Medicine

plosmedicine.org

Requests from Editors:

1. Please define abbreviations in tables e.g. PLHIV.

2. Please indicate in the figure caption the meaning of the bars and whiskers in Figure S4.

3. Please provide your MOOSE checklist and complete it with paragraph numbers per section (e.g. "Methods, paragraph 1").

4. The survey flowchart can be included in the main paper.

5. To help us extend the reach of your research, please provide any Twitter handle(s) that would be appropriate to tag, including your own, your coauthors’, your institution, funder, or lab.

Comments from the Academic Editor:

I would ask the authors to comment in a bit more detail the issue of 'paying' for sex as opposed to 'gifts etc'. There are some sentences relating to that in the limitation, which are fine, but for me the issue would be the overall rather low prevalence of men paying for sex (only 8% overall), which is lower than would be anticipated in terms of 'bought' sex (irrespective of whether it was money or gifts), especially in South/southern Africa. The authors may like to comment on the prevalence of paying for sex, by region. In South/southern Africa the concept of 'sugar daddy' is relatively common (and may involve more than one recipient at a time), but usually does not involve money directly and it is likely that this would not have been reported as 'paying for sex with a sex worker'. I appreciate that the authors are limited by the data available but could still comment on that in the limitations, and the generalisability of the conclusion.

Comments from Reviewers:

Reviewer #1: The authors have addressed all my points

Michael Dewey

[LINK]

---

## [Editor Report · Decision Letter 3]

4 Nov 2021

Dear Dr Maheu-Giroux, 

On behalf of my colleagues and the Academic Editor, Dr. Marie-Louise Newell, I am pleased to inform you that we have agreed to publish your manuscript "HIV prevalence, population sizes, and HIV prevention among men who paid for sex in sub-Saharan Africa (2000-2020): a meta-analysis of 87 population-based surveys" (PMEDICINE-D-21-02581R3) in PLOS Medicine.

PUBLICATION SCHEDULE

Given our busy publication schedule for the remainder of 2021, we are planning to publish your paper in early January 2022 (the exact date will be communicated to you once confirmed).

PRESS

Sincerely, 

Beryne Odeny 

PLOS Medicine